# The Brain’s Glymphatic System: Drawing New Perspectives in Neuroscience

**DOI:** 10.3390/brainsci13071005

**Published:** 2023-06-28

**Authors:** Alexandru Vlad Ciurea, Aurel George Mohan, Razvan-Adrian Covache-Busuioc, Horia Petre Costin, Vicentiu Mircea Saceleanu

**Affiliations:** 1Neurosurgery Department, “Carol Davila” University of Medicine and Pharmacy, 020021 Bucharest, Romania; prof.avciurea@gmail.com (A.V.C.); razvan-adrian.covache-busuioc0720@stud.umfcd.ro (R.-A.C.-B.); horiacostin2001@yahoo.com (H.P.C.); 2Neurosurgery Department, Sanador Clinical Hospital, 010991 Bucharest, Romania; 3Department of Neurosurgery, Bihor County Emergency Clinical Hospital, 410167 Oradea, Romania; 4Department of Neurosurgery, Faculty of Medicine, Oradea University, 410610 Oradea, Romania; 5Neurosurgery Department, Sibiu County Emergency Hospital, 550245 Sibiu, Romania; vicentiu.saceleanu@gmail.com; 6Department of Neurosurgery, “Lucian Blaga” University of Medicine, 550024 Sibiu, Romania

**Keywords:** glymphatic system, brain physiology, astrocytes, AQP4, Alzheimer’s disease, circadian rhythm, migraine aura

## Abstract

This paper delves into the intricate structure and functionality of the brain’s glymphatic system, bringing forth new dimensions in its neuroscientific understanding. This paper commences by exploring the cerebrospinal fluid (CSF)—its localization, production, and pivotal role within the central nervous system, acting as a cushion and vehicle for nutrient distribution and waste elimination. We then transition into an in-depth study of the morphophysiological aspects of the glymphatic system, a recent discovery revolutionizing the perception of waste clearance from the brain, highlighting its lymphatic-like characteristics and remarkable operations. This paper subsequently emphasizes the glymphatic system’s potential implications in Alzheimer’s disease (AD), discussing the connection between inefficient glymphatic clearance and AD pathogenesis. This review also elucidates the intriguing interplay between the glymphatic system and the circadian rhythm, illustrating the optimal functioning of glymphatic clearance during sleep. Lastly, we underscore the hitherto underappreciated involvement of the glymphatic system in the tumoral microenvironment, potentially impacting tumor growth and progression. This comprehensive paper accentuates the glymphatic system’s pivotal role in multiple domains, fostering an understanding of the brain’s waste clearance mechanisms and offering avenues for further research into neuropathological conditions.

## 1. Introduction

The lymphatic system encompasses an intricate assembly of cells, tissues, and organs, which collectively operate in a highly organized manner to meticulously regulate the internal fluid environment of the body. This system consists of an extensive network of endothelial vessels, which function to collect the extracellular fluid and prevent its accumulation that would otherwise result in lymphedema. The lymph is funneled through the thoracic duct before draining into the subclavian vein.

In contrast to lymphatic capillaries, lymphatic vessels, through the presence of a smooth muscular layer and valves, facilitate a unidirectional flow of lymph through the lymph nodes [1]. Within the lymph nodes, the ultrafiltrate of blood plasma gathered from the extracellular matrix undergoes constant filtration and monitoring by aggregations of T and B cells, which are vital components of the immune system.

Lymphocytes, along with ancillary cells such as monocytes, macrophages, and granulocytes, comprise the central cellular components of the lymphatic system. These cells are categorized into primary lymphoid organs (bone marrow and thymus) and secondary lymphoid organs (spleen and lymph nodes) [2].

Furthermore, the lymphatic system plays an indispensable role in the immune response owing to the extensive network of lymphatic vessels which serve to detoxify tissues by draining extracellular fluids.

Conversely, the central nervous system (CNS) is characterized by an intricate structure comprising white and gray matter. Both structures incorporate glial cells, which primarily serve a supportive function, particularly abundant in white matter. Microglia, specialized macrophages in the CNS, are instrumental in the immune response within the brain, as they facilitate the elimination of antigens and certain metabolites. Additionally, astrocytes, star-shaped cells that are the most abundant in the human brain, play critical roles in the blood–brain barrier, nutrient transport from capillaries to neurons, and reparative processes following brain injury [3].

It is noteworthy that the CNS, despite being among the most metabolically active systems in the body, accounting for approximately 20% of the body’s energy expenditure, lacks a traditional lymphatic system for waste removal [4].

## 2. CSF—Localization, Production, Role

The CNS presents in its structure four cavities named ventricles filled with an ultrafiltered blood plasma called cerebrospinal fluid (CSF). Inside each of the four ventricles, a structure called the choroid plexus, consisting of modified ependymal cells, takes part into the secretion of the CSF that travels through the subarachnoid space, in order to end up in the sagittal sinus, by means of the arachnoid granulations [5]. Since the CSF is then secreted in the ventricular system, mostly by the 2 lateral ventricles, clear surveillance of the components of this liquid is important, and this is where the action of the macrophages, the astrocytes and the tight junctions between the choroid ependymal cells, which creates a barrier between the fenestrated capillary and the CSF system, factors in [6].

The difference between the CSF and blood plasma is that the concentration of sodium, chloride and magnesium is higher in the CSF while the concentration of potassium and calcium is higher in the blood plasma. Moreover, the CSF only contains a considerably small quantity of proteins and cells [7]. The first role of the CSF is shock absorption, preventing the brain from hitting the skull, therefore reducing the probability of an injury [8]. Secondly, the CSF can be regarded as buoyant, since it is capable of reducing the weight of the brain from 1400 to 50 g [9]. Thus, the possibility of an injury is diminished for both the CNS and the vasculature, which could be constricted by the weight. However, in recent years, a role of the CSF which is not so clear yet was discovered. It was shown that the CSF plays a role in maintaining the homeostasis of the brain’s interstitial space, in order to maintain the proper functioning of the neurons [10].

Since this role is attributed in the body to the ultrafiltered plasma, representing waste removal fluid, via the lymph and later via the venous system, the brain does not allow this production due to the blood–brain barrier (BBB). For this reason, it becomes more and more clear that the CSF replaces the lymph. As a comparison between the lymphatic system and the glymphatic one, both the ultrafiltrate of plasma and the CSF can occur at the arterial bed of the microvasculature. However, the former will drain in the lymphatic system after clearing the interstitial space, while the other will end up in the paravenous space [11]. However, even if both the CSF and the lymphatic fluid drain through the lymphatic system and ganglia after that, the former can take two different paths. Firstly, it can drain in the venous system indirectly through the lymphatic network, through either the spinal nerve roots or the cribriform plate. Secondly, the CSF can end up directly in the venous sinuses using the arachnoid villi (arachnoid granulations), which help the efflux of the CSF from the subarachnoidian space [12].

## 3. The Morphophysiological Aspects of the Glymphatic System

It appears that the glymphatic system of the CNS plays a crucial role in maintaining the homeostasis of the brain parenchyma. Nonetheless, one important factor for the entire CNS is regarded as the perivascular space (PVS), which represents the space between the astrocytes end-foot and the smooth muscle and vascular endothelium of the brain vasculature. Further on, the CSF from the PVS will drain through the dural lymphatic vessels to the cervical lymphatic nodes [13]. 

Both the arteriole and the astrocytes play a key role in determining the influxes of the CSF through the brain parenchyma, since the vasodilatation or vasoconstriction of the arterial end, as well as the swelling of the astrocyte end-foot, can alter the bulk flow of the CSF to the paravenous space.

According to a study conducted by Mestre et al., (2020) [14] that reviewed the post-ischemic edema showed that due to the spreading depolarization that leads to a vasoconstriction of the arterioles in the focal cerebral infarct, the PVS will get enlarged. This action is responsible for an increased volume of CSF, which will contribute to the swelling of the tissue, due to the gradient change, therefore doubling the speed of the CSF inflow into the tissues.

However, we should also take into account the aquaporin-4 channel (Aqp-4) found with a higher prevalence in the astrocyte end-foot that makes the link between the para-arterial space and paravenous space. Thus, the CSF can pass freely through the BBB, making its way into the interstitial place [15]. Studies showed that the inactivation of the Aqp-4 channel in the brain of the mouse, leads to an altered water balance. This abnormality has consequences such as brain edema, hydrocephalus and stroke [16]. Moreover, studies have shown that the impairment of the glymphatic system may have much more repercussions than expected.

A study conducted by Aaron J. Schain et al. [17], showed that the CSD (cortical spreading depression), which is a slowly propagating wave due to an altered brain activity regarded as responsible for inducing the migraine aura, creating a severe disruption of the ionic homeostasis. It increases the concentrations of K+, neurotransmitters such as Glutamate, as well as the synthesis of inflammatory factors such as NO synthetase and COX-2 in the brain parenchyma. All of these changes in the homeostasis lead to a sudden vasodilation, decreasing the PVS and therefore reducing the glymphatic flow, as well as contributing to the pathophysiological aspects of the migraine aura.

This discovery leads us to another very important role of the glymphatic system: the protection of the brain against the β-amyloid, the underlying protein of Alzheimer’s disease.

The existence of the Aqp-4 in the structure of astroglial end-feet plays an important role in linking the para-arterial space, the interstitial space and the paravenous space. Therefore, besides maintaining a firm homeostasis of the brain parenchyma, it also takes part in clearance of the metabolic wastes produced by the brain (Figure 1) [18].

## 4. The Glymphatic System in AD 

Alzheimer’s disease (AD), the most prevalent neurodegenerative dementia, affects over 50 million individuals globally, making it a serious public health challenge [19,20]. AD is irreversible disease marked by neurodegeneration caused by harmful buildups of extracellular amyloid plaques and intracellular neurofibrillary tangles composed of hyperphosphorylated tau protein that results in neuronal damage which eventually leads to cognitive deterioration as well as personality and behavioral changes [21].

The traditional amyloid cascade hypothesis postulates that amyloid-b accumulation is among the initial factors leading to Alzheimer’s disease (AD) and that its further progression, including neurofibrillary tangle formation containing tau protein, results from an imbalance between Aβ production and clearance [22,23]. Other neurodegenerative disorders have similar proteins accumulating both inside and outside cells such as Lewy Bodies or neurites found in Parkinson’s Disease; such accumulations contribute to neurodegeneration while being implicated in both pathogenesis of both AD and PD, but their exact biological linkage with their biological relationship within their respective glymphatic system remains unknown [24].

The glymphatic system may play an integral role in removing a-synuclein and thus impacting PD progression. A negative correlation has been observed between deposition of a-synuclein and expression of water channel protein AQP4 in brains of patients suffering PD, suggesting glymphatic dysfunction may contribute to protein accumulation (Figure 2) [25]. Clearance of Aβ and tau proteins into cerebrospinal fluid (CSF) serves as the basis for measurement purposes as clinical biomarkers of AD [26,27].

Though Aβ is widely believed to travel through the blood–brain barrier (BBB), recent evidence from Nauen and Troncoso’s study [28] indicates otherwise; their discovery of Aβ in human lymph nodes suggests alternative exit routes from central nervous system for Aβ and similar proteins, suggesting that it may also use the lymphatic system as part of its clearance from brain.

Aging, with its altered circadian rhythms and associated sleep deprivation, has long been recognized as a risk factor for neurodegenerative disorders such as Alzheimer’s disease and Parkinson’s disease. Aging also causes efficiency loss between subarachnoidal CSF exchange and brain parenchyma exchange, thus decreasing efficiency between CSF exchange and brain parenchyma exchange [29]. Evidence supporting the decline of glymphatic system function with age comes both from experimental models and patients with neurological conditions [30].

Recent work conducted by McKee and colleagues [31] has examined the relationship between circadian clock function and neurodegeneration. Their studies examined how astrocyte activation caused by deletion of core clock gene Bmal1 affects gene expression, Aβ plaque activation and deposition. Deletion disrupts clock function while inducing cell-autonomous activation in astrocytes.

Studies indicate that any impairment to the glymphatic system’s ability to clear extracellular tau, a protein, may contribute to tau aggregation and neurodegeneration [32]. Furthermore, deficiency of water channel protein AQP4 could exacerbate tau accumulation further and create a vicious cycle between compromised clearance of extracellular tau and accumulation on other cells [32]. Although exact mechanisms by which impaired extracellular tau clearance exacerbates tau-related pathology remain unknown, impaired clearance among PS19 mice deficient for AQP4 could facilitate spread of pathological tau species over other cells, suggesting impaired extracellular tau clearance could promote spread over other cells resulting in pathologically spread of pathological tau species to other cells and organs [33].

Overall, both deletion of AQP4 or its pharmacological inhibition amplifies pathogenic accumulations of Aβ and tau in AD transgenic mouse models. Furthermore, recent genetic research has demonstrated that various SNPs within the AQP4 gene can influence cognitive decline following an AD diagnosis; two are linked with slower decline whereas others such as rs9951307 and rs3875089 may lead to faster cognitive deterioration post diagnosis while two others (rs3763040 and rs3763043) could promote rapid decline [34,35].

Variations in the AQP4 gene have been linked with Aβ accumulation, disease stage progression and cognitive decline, suggesting that it could serve as a biomarker to accurately predict disease burden in those within the AD spectrum. Furthermore, certain SNPs of this gene have been linked with reduced perivascular localization of AQP4 protein among AD patients [34,35].

According to a study conducted by Iliff et al. [36], the soluble β-amyloid clearance is reduced by almost 55% in case of an Aqp-4 null mouse compared to a normal wild-type mouse.

Consequently, both the perturbance of the astrocyte’s vascular end foot that presents the Aqp-4 as well as a decreased glymphatic flow (which can be caused due to the CSD that produces the vasodilation, underlying the physiopathology of the migraine aura) lead to an increased risk of Alzheimer’s disease development [37].

## 5. The Glymphatic System and the Circadian Rhythm

Recent studies demonstrate a clear correlation between sleep disruption, brain glymphatic system dysfunction and Alzheimer’s disease (AD) [38,39,40]. Similar to lymphatic systems found elsewhere, glymphatic systems consist of para-vascular channels within the brain’s blood vessels that carry cerebrospinal fluid (CSF) to capillary beds before penetrating parenchyma tissue where it mixes with interstitial fluid, collects metabolic waste and eventually returns back through paravenous space and then cervical lymphatic vessels for removal [41].

The glymphatic system plays an essential role in clearing away neurotoxic substances such as Aβ from central nervous system parenchyma [42,43]. Studies have demonstrated that over half of Aβ can be eliminated through this mechanism. Sleep can have an interesting influence on the functioning of the glymphatic system. When sleeping naturally, brain interstitial space vastly expands compared to wakefulness—possibly as a result of astroglial cell shrinkage [38,44]. An expansion of extracellular space accelerates clearance processes; for instance, mice demonstrated that Aβ clearance during sleep was twice as fast compared to wakefulness [45]. Another study demonstrated that clearance through the glymphatic system also depends on body posture—with lateral positions commonly adopted during sleeping being most efficient for clearing [46].

As Aβ clearance is impaired in both early and late stages of AD [47], it is plausible to propose that there may be a connection between impaired glymphatic system function and AD. Studies in animals and humans have documented diurnal oscillation of Aβ level in brain interstitial fluid; endogenous neuronal activity influences regional Aβ concentration [48]; decreased neuronal activity during certain sleep stages may cause this fluctuation, potentially contributing to both impaired clearance as well as disturbances in Aβ production caused by disturbed slow wave sleep patterns [49]. Thus, altered sleep quality may contribute to both its onset and progression through impaired clearance as well as disruptions of Aβ production caused by reduced endogenous neuronal activity influencing regional concentration of Aβ [49,50].

Sleep stages, specifically slow wave sleep, has been found to influence Aβ 42 levels in the cerebrospinal fluid (CSF). A study involving 36 cognitively normal and elderly subjects discovered an inverse relationship between CSF Aβ 42 levels and slow wave sleep duration, percentage of total sleep time spent sleeping on slow waves, frontal EEG leads activity while sleeping, local Aβ accumulation during low-frequency-range sleep time and reduced slow wave activity during these hours [51], suggesting there may be an association between decreased Aβ clearance or production and deficiency and slow wave sleep deficiency and reduced clearance/production/clearer production/clearance/production and slow wave sleep deficiency and slow wave sleep deficiency [52].

Nonetheless, the glymphatic system seems to operate in accordance with the circadian rhythm. The role of this rhythm is to create different neural activity peaks, in order to maintain a good balance between sleep and active phase [45].

The role of the glymphatic system is to synchronize the entire rhythmicity, since the suprachiasmatic nucleus of the hypothalamus which represents the main pacemaker is connected to the CSF. Moreover, other neurotransmitters such as VIP (Vasoactive Intestinal Peptide) and AVP (Arginine Vasopressin) are transported through the CSF. Therefore, the entire glymphatic system could be responsible for through its bulk flow to coordinate the neurons, the physiological mechanisms of the brain and in the end the entire behavior of the human [29].

## 6. Glymphatic System Involvement in the Tumoral Microenvironment

Studies on rodents have provided invaluable insights into the changes to glymphatic function that correlate with the development of gliomas. These animal studies observed a decreased rate of cerebrospinal fluid (CSF) efflux and restructuring of the glymphatic pathway, both indicators that could contribute to brain edema around gliomas [53,54]. Animal studies have considerably broadened our knowledge, yet it is essential that their findings can be directly applied to human glymphatic systems. A variety of noninvasive techniques have been suggested for studying this function in human bodies; structural and diffusion MRI have both been proposed as noninvasive approaches for investigating it [50,55,56].

The ALPS (Analysis Along the Perivascular Space) index, derived from diffusion tensor imaging (DTI) [50], may provide insight into diffusivity within medullary veins at the level of lateral ventricle bodies at medullary vein level and may provide an estimation of human glymphatic function. Variations in ALPS index show correlation with scores on Mini-Mental State Examination tests administered to Alzheimer’s disease patients which could indicate dysfunction, while lower values have also been seen among normal pressure hydrocephalus patients which may suggest impaired functioning system due to delayed clearance of intrathecically administered gadobutrol [54,57]. Given its noninvasive nature and potential as a tool for exploring human glymphatic systems, we decided to utilize the ALPS index for exploring glymphatic function among patients with gliomas. Our research seeks to comprehend any correlations between this aspect of patient care and variables such as tumor and peritumoral brain edema volumes, tumor grades and IDH1 mutation status. 

Traditional theories regarding peritumoral brain edema in gliomas suggest it occurs due to net fluid transport from intravascular compartments into brain interstitial spaces as a result of microvessel proliferation with defective inter-endothelial tight junctions [58], but this does not account for its occurrence with low grade gliomas with intact tight junctions. Research demonstrated an inverse relationship between ALPS index and peritumoral brain edema volume, suggesting brain edema associated with intra-axial tumors may also be affected by malfunction in the glymphatic system [59,60].

Recent research suggests that changes to the supporting structures of the blood–brain barrier, including astrocytes, pericytes, and microglial cells, could contribute to fluid entering the interstitium of the brain. Astrocyte coverage of brain microvessels appears to be an impediment to water movement, and AQP4 channels located on astrocytic foot processes could play an essential part in creating peritumoral brain edema. Evidence showing a correlation between peritumoral brain edema and elevated AQP4 expression levels in human gliomas astrocytes indicates that increased expression could play a key role in its pathogenesis [61,62]. As AQP4 water channels form part of the glymphatic system, we hypothesize that there may be a correlation between ALPS index values and expression of AQP4 water channels and expression levels in gliomas. Further research needs to confirm its use as an imaging marker of AQP4 expression in these tumors.

Changes in glymphatic function due to glioma growth have only been explored through limited animal studies. An investigation using an orthotopic xenograft glioma model demonstrated how tumor growth led to reduced CSF flow into extracranial spaces, leading to dysfunction and lower glymphatic flow overall. It was found that IDH1 wild-type gliomas, as measured by ALPS index scores, demonstrated significantly lower glymphatic performance than their IDH1 mutant counterparts; no human studies have reported any correlation between changes in glymphatic function and IDH1 mutation status or gene status for any human studies either way [53,63].

## 7. Conclusions

In conclusion, the glymphatic system of the brain is similar to the lymphatic systems of the entire body, whereas the role and mechanism of it represents critical discoveries regarding the deeper understanding of the human brain. It seems that its role is not so easy to comprehend, since it has implications in both physiological and pathological aspects of the brain, regarded as complex pathways that must be discovered.

Given AQP4’s crucial role in the glymphatic system and potential implications for Alzheimer’s disease, future research should include this water channel. It has been suggested as a promising therapeutic target for AD due to its effects on Aβ and tau clearance as well as neuronal function improvement, making it highly relevant in relation to both aging and neurodegeneration. AQP4, naturally upregulated with age but mislocalized in AD brains, could offer therapeutic potential by being modulated [64,65].

The blood–brain barrier (BBB) is highly selective [64], restricting drug passage from the bloodstream into extracellular fluids in the brain. Some conditions are associated with compromised BBBs that allow drug delivery; this is not the case in early-stage AD, where anti-amyloid agents such as bapineuzumab have failed clinical trials due to being unable to access amyloid plaques directly [66]; any drug designed to effectively target AQP4 and be relevant in treating AD must therefore be capable of crossing an intact BBB in order to be effective against its progression.

Even though the brain’s CSF drainage function is an integrated system involving various compartments, the glymphatic system and AQP4 could serve as intervention targets in neurodegenerative diseases. Enhancing its function and efficiency could help delay or prevent protein accumulation in the brain; additionally, this mechanism could play a critical role in clearing away tau from circulation; special attention should be paid to Aβ-independent regulators of tau, such as the glymphatic system, when investigating neurodegenerative tauopathies [67].

Due to the importance of decreased molecular clearance from CSF to blood, which contributes to neurological diseases, direct measurement of CSF to blood clearance on an individual basis has become a method for personalized intrathecal drug administration in the central nervous system [68].

Moreover, we delve deeper into the intriguing relationship between the glymphatic system and migraine aura, offering new insight into its pathophysiology. This discovery might open up possibilities for research and treatments of this debilitating condition.

Furthermore, the interaction between the glymphatic system and circadian rhythms has been explored, drawing attention to their potential impact on brain health and function as well as sleep cycles and biological rhythms. Such findings could have significant ramifications for understanding and treating various neurological disorders.

Finally, it is vital to identify neuroimaging markers to detect changes in the glymphatic system, and noninvasive techniques could serve as effective tools to detect dysfunction in this area and become new potential biomarkers in AD.

## Figures and Tables

**Figure 1 brainsci-13-01005-f001:**
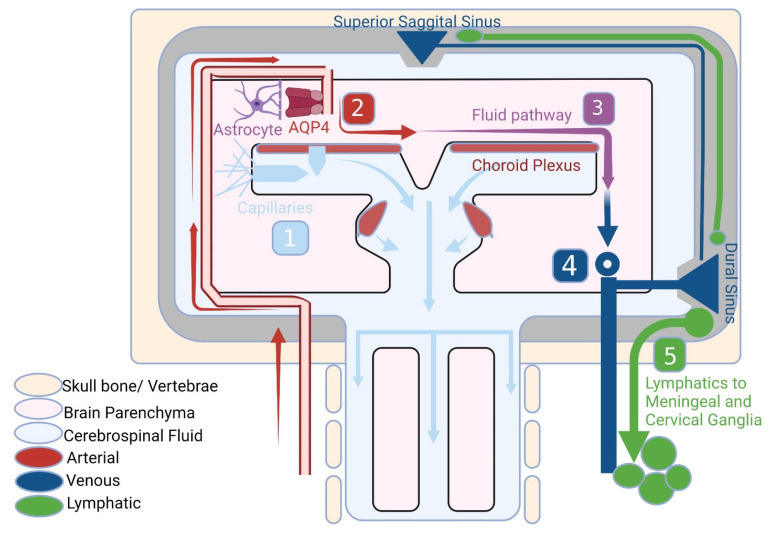
The pathway for fluid transport can be separated into five distinct sections: (1) the production of cerebrospinal fluid (CSF) by the choroid plexus, potentially supplemented by extrachoroidal sources such as capillary influx and metabolic water production; (2) the pulsation of arterial walls which drives CSF deep into the brain along perivascular spaces; (3) the CSF is carried into the brain parenchyma through AQP4 water channels and then dispersed throughout the neuropil; (4) interstitial fluid (ISF) and CSF mix, accumulating in the peri-venous space and subsequently draining from the brain through (5) meningeal and cervical lymphatic vessels, as well as cranial and spinal nerves.

**Figure 2 brainsci-13-01005-f002:**
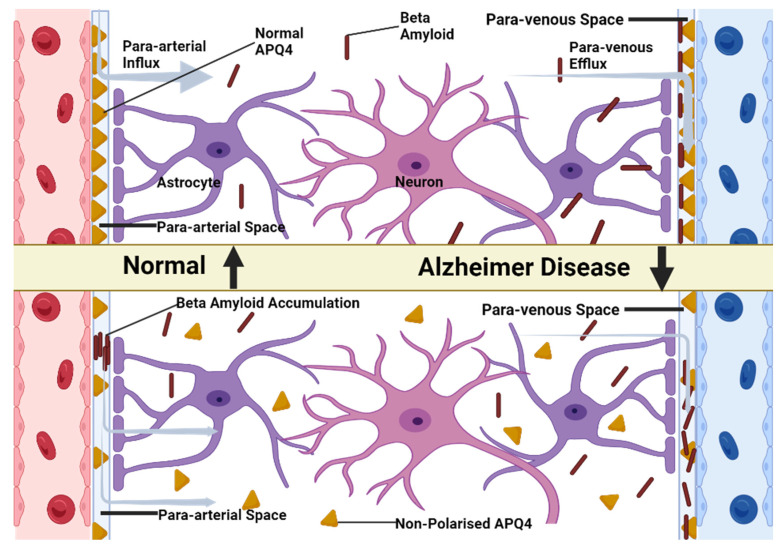
AQP4 aids in the elimination of waste via the glymphatic system. Under normal conditions, AQP4 is primarily located on the end-feet of astrocytes, which is referred to as polarized AQP4 expression. The cerebrospinal fluid (CSF) flows into the brain tissue through the para-arterial system, or para-arterial influx, and then exits into the veins, a process known as para-arterial efflux. However, as we age, and particularly in pathological conditions, the polarization of AQP4 diminishes, leading to an increased presence of AQP4 on parenchymal processes. This phenomenon, known as AQP4 depolarization, hampers the glymphatic system’s ability to effectively clear waste, such as beta-amyloid.

## Data Availability

All data is available on PubMed.

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
