# Peer review of "The Brain’s Glymphatic System: Drawing New Perspectives in Neuroscience"

_brainsci, 2023, doi:10.3390/brainsci13071005_

Round 1

Reviewer 1 Report

This review of the Glymphatich system involved in the development of many pathologies, including Alzheimer disease. It is a basic review that helps us to better understand the topic.

It was easy to read, fluid and interesting. There are still some doubts that remain for me, and I have missed seeing more findings regarding Alzheimer's disease (AD). In the meantime, I always expected to see the relationship of the Glymphatich system with migraines and Alzheimer's disease. And it was only commented on towards the end of the review.

Furthermore, for a review, most references used are older than 5 years, some older than 10 years. Please include some news references.
The pathway scheme is very clear and auto explicative.

Author Response

Reviewer 1: “This review of the Glymphatich system involved in the development of many pathologies, including Alzheimer disease. It is a basic review that helps us to better understand the topic.

It was easy to read, fluid and interesting. There are still some doubts that remain for me, and I have missed seeing more findings regarding Alzheimer's disease (AD). In the meantime, I always expected to see the relationship of the Glymphatich system with migraines and Alzheimer's disease. And it was only commented on towards the end of the review.

Furthermore, for a review, most references used are older than 5 years, some older than 10 years. Please include some news references.
The pathway scheme is very clear and auto explicative.”

Response: Thank you very much for your expertise, assistance and comments!

We seriously took everything into considerations and adressed all of your comments!

 “I have missed seeing more findings regarding Alzheimer's disease (AD). In the meantime, I always expected to see the relationship of the Glymphatich system with migraines and Alzheimer's disease.” – We have added a lot of new data regardind Alzheimer disease, Migraina aura, Circadian rhytm and Tumoral Microenviroments related to the Glymphatic System in the revised version!.

“Furthermore, for a review, most references used are older than 5 years, some older than 10 years. Please include some news references.
The pathway scheme is very clear and auto explicative” -  We have added more than 40 new citations, including the most recent research regarding the topic. Moreover, we have added a new figure to better understand the pathological mechanism of Glymphatic System in Alzheimer disease.

Overall, thank you for your expertise, time and suggestions!!

Reviewer 2 Report

This topic is very current, as several recent papers talked about it. Some points need to be revised:

- "However, a great paradox occurs when we speak about the central nervous system (CNS) waste removal system" What do authors mean? Revise it.

- It is not clear what is the purpose of this paper? Revise it.

- Is this a review? If yes, add some inclusion or exclusion criteria, or a PRISMA Flow Diagram.

- The author seems to describe only the pathophysiology of this system. It is also better to talk also about some brain diseases, such as tumors and lymphoid cells involved in the tumor microenvironment in a new paragraph. Consider these refs: --  Macrophages in Recurrent Glioblastoma as a Prognostic Factor in the Synergistic System of the Tumor Microenvironment. Neurol Int. 2023 Apr 23;15(2):595-608  --  Current status of intratumoral therapy for glioblastoma. J Neurooncol. 2015 Oct;125

- perhaps the authors could add more pictures to better describe what they found new

- conclusion must be revised. It seem that this review add nothing new to the literature. What did the authors find new when writing this article?

Minor editing of English language required

Author Response

Reviewer 2: This topic is very current, as several recent papers talked about it. Some points need to be revised:

- "However, a great paradox occurs when we speak about the central nervous system (CNS) waste removal system" What do authors mean? Revise it.

- It is not clear what is the purpose of this paper? Revise it.

- Is this a review? If yes, add some inclusion or exclusion criteria, or a PRISMA Flow Diagram.

- The author seems to describe only the pathophysiology of this system. It is also better to talk also about some brain diseases, such as tumors and lymphoid cells involved in the tumor microenvironment in a new paragraph. Consider these refs: --  Macrophages in Recurrent Glioblastoma as a Prognostic Factor in the Synergistic System of the Tumor Microenvironment. Neurol Int. 2023 Apr 23;15(2):595-608  --  Current status of intratumoral therapy for glioblastoma. J Neurooncol. 2015 Oct;125

- perhaps the authors could add more pictures to better describe what they found new

- conclusion must be revised. It seem that this review add nothing new to the literature. What did the authors find new when writing this article?

Response:

Thank you very much for your expertise, assistance and comments!

We seriously took everything into considerations and adressed all of your comments!

  1. - "However, a great paradox occurs when we speak about the central nervous system (CNS) waste removal system" What do authors mean? Revise it.

- It is not clear what is the purpose of this paper? Revise it.

We have revised and addressed this comments accordingly!

  1. - Is this a review? If yes, add some inclusion or exclusion criteria, or a PRISMA Flow Diagram.

No, it is not a review; it is an Opinion type article.

  1. - The author seems to describe only the pathophysiology of this system. It is also better to talk also about some brain diseases, such as tumors and lymphoid cells involved in the tumor microenvironment in a new paragraph. Consider these refs: --  Macrophages in Recurrent Glioblastoma as a Prognostic Factor in the Synergistic System of the Tumor Microenvironment. Neurol Int. 2023 Apr 23;15(2):595-608  --  Current status of intratumoral therapy for glioblastoma. J Neurooncol. 2015 Oct;125

Thank you very much for this suggestion! We have added a new section about Tumoral Microenvironment related to Glymphatic System.

  1. - perhaps the authors could add more pictures to better describe what they found new

We have added another figure in order to better explain the pathology of Alzheimer’s Disease regarding the Glymphatic System.

  1. - conclusion must be revised. It seem that this review add nothing new to the literature. What did the authors find new when writing this article?

We have revised the conclusion, as well as the whole manuscript. We added a lot more relevant and actual information, as well as bringing novelty by linking Alzheimer’s disease, Circadian rhytm, Migraine aura and Tumoral Microenviroments to the Glymphatic system as you can see in the revised manuscript.

We hope the Major Changes we have succeeded addressing all of  your comments and concerns!

Reviewer 3 Report

Ciurea and co-authors prepared a brief overview of the anatomy and physiology of the brain's glymphatic system. See comments below.

(1) The provided text and accompanying Figure illustration are informative. However, there is a major concern regarding the organization and breadth of the article. Each point of information does not necessarily flow in a logical manner and it is very unusual to see each sentence as its own paragraph. Transitional statements would help to narrate the purpose of each sentence and perhaps some sentences can be combined as well. 

(2) A strong recommendation is to declare goals/objectives of the article up front in the Introduction (what do the authors want the reader to learn?), then carefully and thoroughly present the cellular and molecular anatomy of the glymphatic system, and then its functional mechanisms (the Figure is helpful!), and conclude with outstanding/controversial questions that remain in the field of study.  

The article requires better organization as with recommendations suggested above.

Author Response

Reviewer 3

“Ciurea and co-authors prepared a brief overview of the anatomy and physiology of the brain's glymphatic system. See comments below.

(1) The provided text and accompanying Figure illustration are informative. However, there is a major concern regarding the organization and breadth of the article. Each point of information does not necessarily flow in a logical manner and it is very unusual to see each sentence as its own paragraph. Transitional statements would help to narrate the purpose of each sentence and perhaps some sentences can be combined as well. 

(2) A strong recommendation is to declare goals/objectives of the article up front in the Introduction (what do the authors want the reader to learn?), then carefully and thoroughly present the cellular and molecular anatomy of the glymphatic system, and then its functional mechanisms (the Figure is helpful!), and conclude with outstanding/controversial questions that remain in the field of study.”

Response:

  1. We have completely reorganized the whole manuscript and made the necessary modification in order for the text to flow in a logical manner. Moreover, we have added more actual and relevant research regarding the topic. We hope that Major Changes we have made address this concerns.
  2. We have followed your suggestion, adding objectives in the abstract and then delving deeply into the molecular aspects of various pathologies. Moreover, we have added another figure and a graphical abstract! We have fully reworked the conclusions and made them more relevant by addressing controversial questions that remain in the field of study.

Overall, thank you for your expertise, time and suggestions!! – It really helped bringing more value to this manuscript.

We hope the Major Changes we have succeeded addressing all of  your comments and concerns!

Round 2

Reviewer 2 Report

ok

ok